# Temporal artery temperature measurements versus bladder temperature in critically ill patients, a prospective observational study

**Eline G. M. Cox**[1]*, **Willem Dieperink**[1,2], **Renske Wiersema**[1], **Frank Doesburg**[1], **Ingeborg C. van der Meulen**[1,2], **Wolter Paans**[1,2]

**1** Department of Critical Care, University Medical Center Groningen, University of Groningen, Groningen, The Netherlands, **2** Research Group Nursing Diagnostics, Hanze University of Applied Sciences, Groningen, The Netherlands

* e.g.m.cox@umcg.nl

## Abstract

### Purpose

Accurate measurement of body temperature is important for the timely detection of fever or hypothermia in critically ill patients. In this prospective study, we evaluated whether the agreement between temperature measurements obtained with TAT (test method) and bladder catheter-derived temperature measurements (BT; reference method) is sufficient for clinical practice in critically ill patients.

### Methods

Patients acutely admitted to the Intensive Care Unit were included. After BT was recorded TAT measurements were performed by two independent researchers (TAT$_1$; TAT$_2$). The agreement between TAT and BT was assessed using Bland-Altman plots. Clinical acceptable limits of agreement (LOA) were defined a priori (<0.5˚C). Subgroup analysis was performed in patients receiving norepinephrine.

### Results

In total, 90 critically ill patients (64 males; mean age 62 years) were included. The observed mean difference (TAT-BT; ±SD, 95% LOA) between TAT and BT was 0.12˚C (-1.08˚C to +1.32˚C) for TAT$_1$ and 0.14˚C (-1.05˚C to +1.33˚C) for TAT$_2$. 36% (TAT$_1$) and 42% (TAT$_2$) of all paired measurements failed to meet the acceptable LOA of 0.5˚C. Subgroup analysis showed that when patients were receiving intravenous norepinephrine, the measurements of the test method deviated more from the reference method (p = NS).

### Conclusion

The TAT is not sufficient for clinical practice in critically ill adults.

**Data Availability Statement:** All relevant data are within the manuscript and its Supporting Information files.

**Funding:** The author(s) received no specific funding for this work.

**Competing interests:** The authors have declared that no competing interests exist.

**Abbreviations:** ICU, Intensive Care Unit; LOA, Limits Of Agreement; TAT, Temporal Artery Thermometer; BT, Bladder-catheter derived temperature; UMCG, University Medical Center Groningen; SOCCS, Simple Observational Critical Care Studies; SICS, Simple Intensive Care Studies; SD, Standard Deviation.

## Introduction

Accurate measurement of body temperature is important for the timely detection of fever or hypothermia in patients admitted to the Intensive Care Unit (ICU). Measurements of body temperature are frequently used as a trigger or target for interventions and treatment decisions, especially in patients acutely admitted to the ICU. However, there is no uniform method for non-invasive intermittent measurement of body core temperature in the ICU.

Various devices are used worldwide to measure body temperature at different anatomical locations. Continuous body temperature can be measured in the ICU by the pulmonary artery (PA) thermometer or bladder catheter with thermistor (BT). The PA thermometer is considered to be the gold standard for temperature measurement in critically ill patients since it has been shown to measure the closest to the temperature in the high internal jugular vein [1]. However, this method is associated with a risk for adverse events and is not generally used in the ICU [2]. In contrast, almost all ICU patients have a BT for urine output monitoring. This invasive method has a reliable concordance with the PA catheter temperature measurements and is therefore often considered as the reference method [3, 4].

There are different methods to monitor intermittent, non-invasive, body temperature which are accompanied by different risks and benefits [5, 6]. A relatively new method is the temporal artery (forehead) thermometer (TAT). This method is widely implemented and has replaced the tympanic thermometer in many hospitals. The TAT was introduced to be more accurate, sustainable, easier to use with training and cheaper than other non-invasive instruments for temperature monitoring. However, since the introduction of this instrument, nurses and physicians have had doubts about the accuracy of the measurements. Several prospective studies showed an acceptable agreement between TAT and peripheral thermometers in critically ill patients while other studies showed an unacceptable agreement [5, 7, 8].

Vasoactive medication (i.e. norepinephrine) theoretically is an important factor in TAT measurement as it induces peripheral vasoconstriction and thus may influence local temperature measurement [9]. To our knowledge there are no studies which have assessed this specific finding prospectively in a group of critically ill adults.

As conflicting data exist on the accuracy of TAT in critically ill patients, the aim of this study was to evaluate whether the agreement between body temperature measurements obtained with non-invasive TAT (test method) and BT-derived measurements (reference method) is sufficient for clinical practice in critically ill patients. The secondary objective was to analyze the differences between the two methods separately in patients treated with and without intravenous norepinephrine medication, as norepinephrine can cause peripheral vasoconstriction and therefore potentially influences temperature measurements.

## Methods

### Design and setting

This was a study of the Simple Observational Critical Care Studies (SOCCS) as part of the Simple Intensive Care Studies-II (SICS-II), a prospective observational study designed to evaluate the diagnostic and prognostic value of combinations of clinical examination and hemodynamic variables in critically ill patients (NCT03553069; NCT02912624) [10–12]. The study was conducted in the University Medical Center Groningen (UMCG), a tertiary referral hospital in the Netherlands. The need for consent was waived by the ethics committee as temperature measurement is a daily routine on the ICU (METc 2017/507). Nevertheless, if possible, patients were informed verbally by the researchers and asked for consent before inclusion in

this study. Patients were able to object the use of data obtained for research purposes. None of the included patients objected the use of data for this study.

## Participants and study size

All acutely admitted patients who were 18 years or older with an expected ICU stay of at least 24 hours were eligible for inclusion. Inclusion criteria included the presence of a bladder catheter with thermistor and the accessibility of the locations of the center forehead to hairline and point behind the ear to use the thermometer. Patients were excluded if their ICU admission was planned pre-operatively, if acquiring research data interfered with clinical care due to continuous resuscitation efforts (e.g. mechanical circulatory support), in case of strict isolation, or if informed consent could not be obtained.

## Measurement procedure

All patients were included within the first 48 hours of their ICU admission between 9 am and 3 pm to ensure similar environmental conditions for every patient. Non-invasive body temperature was measured by two independent researchers to minimize the inter-observer variability of the measurements and to evaluate the interobserver difference. Observer 1 ($TAT_1$,) performed the first temperature measurement and within two minutes later, observer 2 ($TAT_2$) performed the second temperature measurement. Measurements were performed using two temporal artery thermometers (Temporal Scanner TAT-5000, Exergen Corp.). To perform the measurements the thermometer was placed on the forehead and then moved along the hairline, after which it was removed from the skin and then place below the earlobe to provide the temperature. Both TAT-5000 instruments were validated and recalibrated before start of the inclusion period and cleaned after every measurement following a predefined protocol supplied by the TAT-5000 manufacturer [13].

## Training

The researchers were two nursing students in their bachelor's degree, trained to conduct a focused body temperature measurement before contributing to the study. Training was given by an expert from the department of Medical Technology of the UMCG and included study theory lessons and practical exercises on healthy individuals, using information from the instrument manufacturer. In addition, the measurements in the first week of inclusion were supervised. The thermometer was used following the available guidelines and after the recommended number training hours according to the manufacturer's advice.

## Variables of interest

Baseline characteristics (age, gender, BMI, APACHE IV score) were collected during a one-time clinical examination in the first 48 hours of patient admission. Reference body temperature was measured invasively by a probe incorporated in a Foley urinary catheter (DeRoyal, Powell, USA) which was already in place. The bladder catheter-derived temperature data were recorded from the display of the bedside monitor IntelliVue MP70 (Philips, Eindhoven, The Netherlands). The dose of intravenous norepinephrine medication, the presence of artificial heating, artificial cooling and moist skin were documented at the time of the temperature measurements. Length of stay was retrieved from the Electronic Health Records after discharge.

## Statistical analysis

Data are presented as means ±standard deviations (SD), or medians with 25th and 75th percentile, or absolute numbers (with percentages). The agreement between the TAT and BT was assessed using Bland-Altman plots, by plotting the mean of the two measurements against their mean difference and 95% LOA (= mean difference (TAT-BT) ± 1.96 × SD of the difference) [14]. The clinically acceptable difference of <0.5˚C between the test and reference method was defined a priori because this can be regarded as a maximum acceptable measurement difference in clinical practice with critically ill patients [15]. An additional analysis was performed using different acceptable margins ranging from 0.1˚C to 1.0˚C. Interobserver agreement was evaluated using the Pearson correlation coefficient. Subgroup analysis was performed in patients receiving norepinephrine. Paired Student's T-test was used to determine whether differences existed among the mean biases of patients receiving norepinephrine or not. P-values of <0.05 were considered statistically significant. All calculations were shown with 95% confidence intervals. For statistical analysis, Microsoft Office Excel 2010 (Microsoft Corp, Redmond, WA, USA) and Stata version 15 (StataCorp, College Station, TX, USA) were used.

## Results

Inclusion for this study started on October 14[th], 2019 and continued until December 13[nd], 2019, during which 161 patients were acutely admitted to the ICU. Thirty-four patients (21%) were missed due to logistic reasons, resulting in 127 patients being assessed for eligibility. A total of 19 patients (15%) were excluded because no bladder catheter with temperature sensor was used, 15 (12%) because there was no access to forehead or ear and 3 (2%) due to traumatic brain injury, leaving 90 patients (71%) for analysis (Fig 1).

Baseline characteristics of all patients are shown in Table 1. The mean age was 62 years and most patients were male (71%). Forty-one patients (46%) received norepinephrine during the

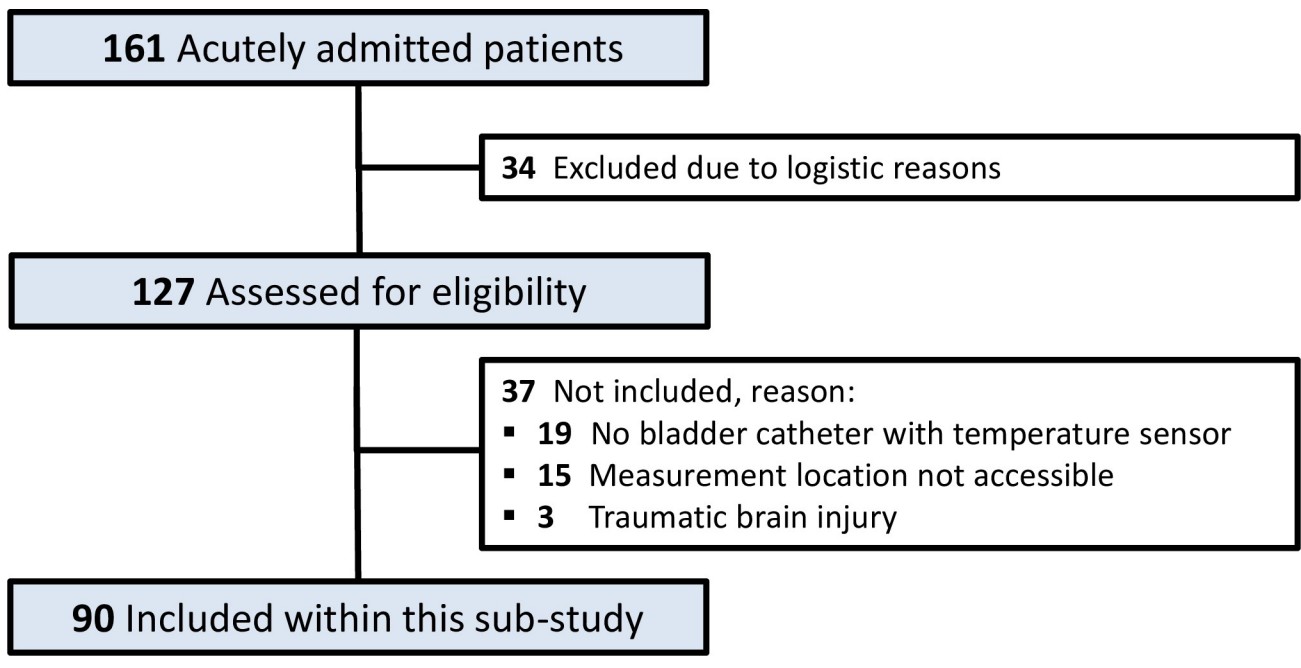

**Fig 1. Flowchart.**

**Table 1. Baseline patient characteristics.**

| Characteristics | *Included in this sub-study* n = 90 |
|---|---|
| Age, years (SD) | 62 (15) |
| Sex, male (%) | 64 (71) |
| BMI, kg/m$^2$ (SD) | 28 (6) |
| APACHE IV, score (SD) | 73 (27) |
| Mean length of stay, days (range) | 5.5 (0–42) |
| Use of vasopressors at inclusion, n (%)* | 41 (46) |
| Use of vasodilators at inclusion, n (%) | 2 (2) |
| Artificial heating, n (%) | 3 (3) |
| Artificial cooling, n (%) | 0 (0) |
| Moist skin, n (%) | 1 (1) |

*All patients who were receiving vasopressors at inclusion, received norepinephrine. BMI: Body Mass Index; APACHE: Acute Physiology and Chronic Health Evaluation score.

temperature measurements. Three patients (3%) received artificial warming, and one patient (1%) was perspiring clearly visible during temperature measurements.

## Temperature measurements

The association between body temperature data obtained non-invasively and invasively by TAT$_1$ and TAT$_2$ is illustrated in scatter plots (Fig 2). BT ranged from 33.8˚C to 38.9˚C, TAT from 35.1˚C to 40.3˚C. Mean BT was 37.1˚C (± 0.81˚C) and both mean TAT$_1$ and mean TAT$_2$ were 37.2˚C (±0.68˚C, ±0.72˚C). TAT Bland-Altman analysis of paired measurements of TAT$_1$ and BT revealed a mean difference (±SD, 95% LOA) of 0.12˚C (± 0.61˚C, -1.08˚C to +1.32˚C) (Fig 3A), indicating that the measurements of TAT$_1$ and TAT$_2$ were overall higher, which implies that the TAT instrument measures a higher temperature than TB. The proportion of TAT$_1$ measurements that deviated 0.5˚C or more was 36% (32 measurements).

Bland-Altman analysis of measurements of TAT$_2$ showed similar results. The mean difference between body core temperature obtained with the BT and the TAT was 0.14˚C (± 0.61˚C, -1.05˚C to +1.33˚C) (Fig 3B). The proportion of TAT$_2$ measurements that deviated 0.5˚C or more was 42% (38 measurements). S1 Table shows the percentage of measurements deviating from different acceptable margins.

**Interobserver agreement.** The distribution and correlation of body core temperature data obtained by TAT$_1$ and TAT$_2$ was illustrated in Fig 4. A strong association was found between TAT$_1$ and TAT$_2$ when measuring body temperature with the TAT (r 0.94; p <0.001).

**Subgroup analysis.** Bland-Altman analysis of paired measurements in patients with (n = 41) or without norepinephrine treatment (n = 49) administration during the examination seemed to show different results. In patients receiving norepinephrine, Bland-Altman analysis showed a mean difference of 0.18˚C (± 0.65˚C, -1.09˚C to +1.45˚C) for TAT$_1$ and 0.20˚C (± 0.63˚C, -1.05˚C to +1.45˚C) for TAT$_2$. For TAT$_1$ 37%, and for TAT$_2$ 32%, of all paired measurements deviated at least 0.5˚C compared to the reference method. However, in the T-test this difference was not significant (p = 0.4).

In patients who did not receive norepinephrine, Bland-Altman analysis showed a mean difference of 0.08˚C (± 0.58˚C, -1.07˚C to +1.22˚C) for TAT$_1$ and 0.09˚C (± 0.09˚C, -1.06˚C to +1.24˚C) for TAT$_2$. 22% (TAT$_1$) and 29% (TAT$_2$) of all measurements deviated 0.5˚C or more compared to the reference method.

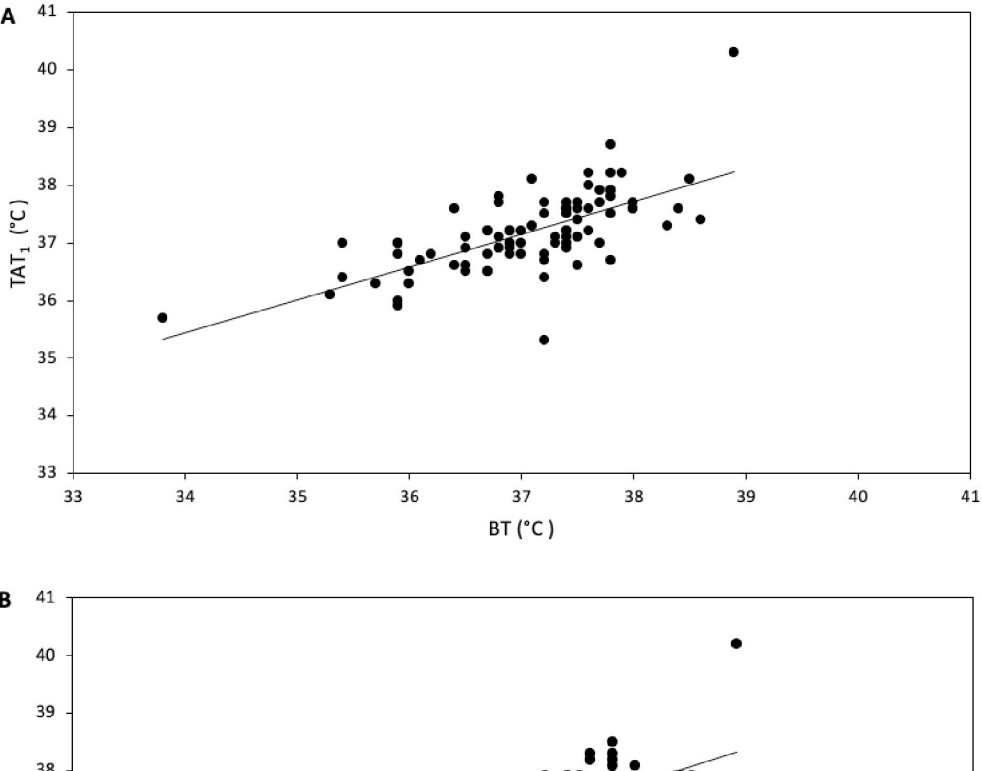

**Fig 2. Scatter plots.** A. Scatter plot for measuring body temperature. Correlation between body temperature data obtained with the temporal artery thermometer by observer 1 (TAT$_1$) and bladder thermometer (BT) is illustrated. B. Scatter plot for measuring body temperature. Correlation between body core temperature data obtained with the temporal artery thermometer by observer 2 (TAT$_2$) and bladder thermometer (BT) is illustrated.

## Discussion

In this prospective observational study, temperature measurement using a TAT showed acceptable agreement, but poor precision compared to BT in critically ill patients. The TAT considerably exceeded the clinical acceptable margin and thus is its use cannot be recommended in critically ill patients.

Our results are in line with most studies comparing peripheral and central body temperature measurements [5, 7, 8, 16]. For example, Kimberger et al. evaluated a TAT in 35 adult patients in a neurosurgical operating room and 35 patients in a neurosurgical ICU and concluded that it was not an adequate substitute for core temperature monitoring [15]. The

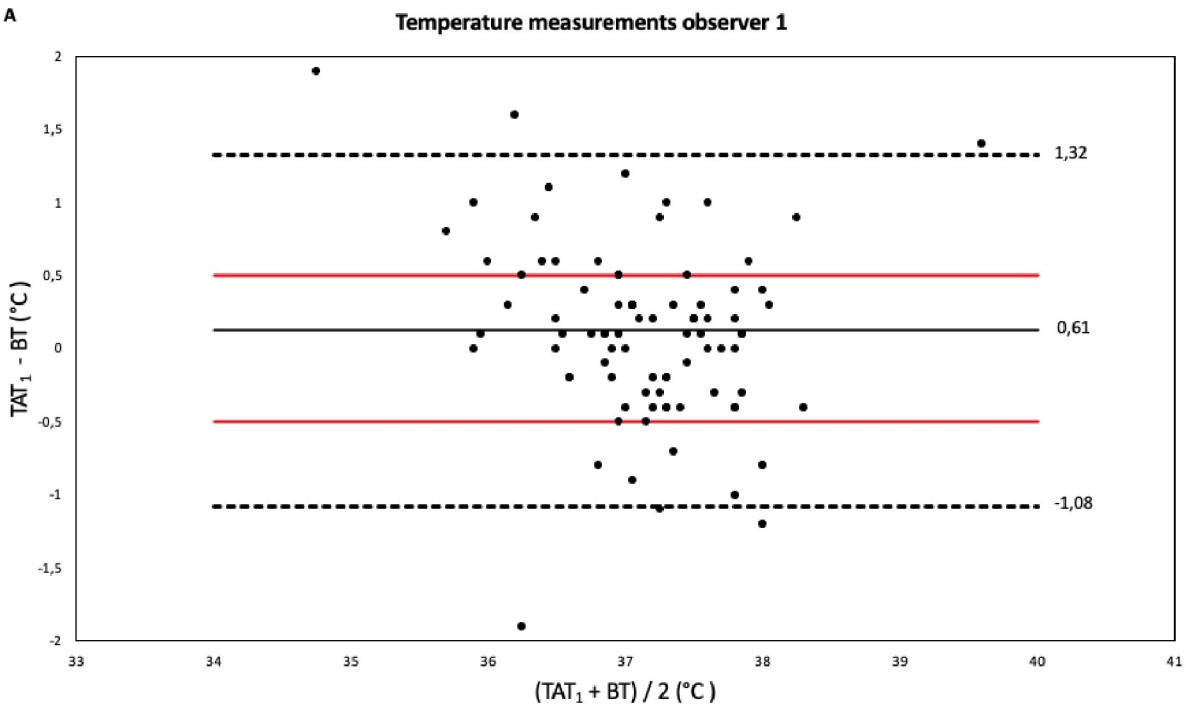

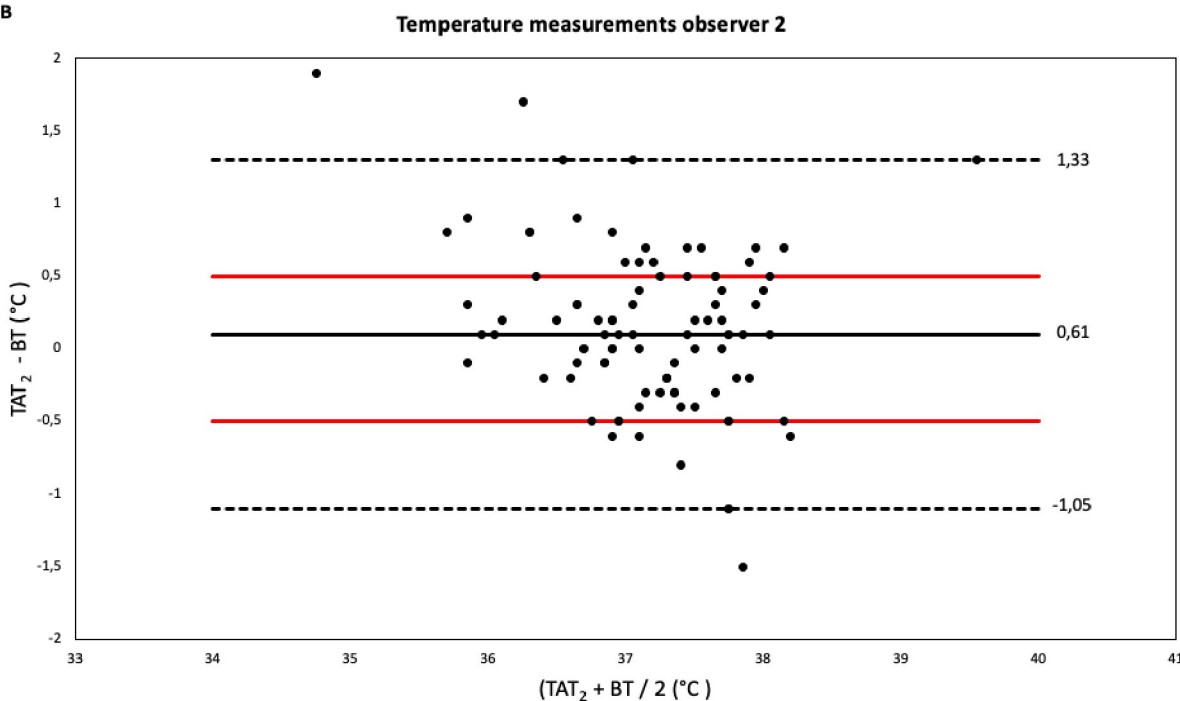

**Fig 3. Bland Altman plots.** A. Bland-Altman plot for body temperature. Comparison of the difference between paired temporal artery (TAT) and bladder temperature (BT) measurements of observer 1 ($TAT_1$) is illustrated. B. Bland-Altman plot for body temperature. Comparison of the difference between paired temporal artery (TAT) and bladder temperature (BT) measurements of observer 2 ($TAT_2$) is illustrated. In each plot, the continuous horizontal line represents the mean difference of the two measurements, and the upper and lower dashed lines represent the 95% limits of agreement. The two red lines represent a tolerance of 0.5˚C.

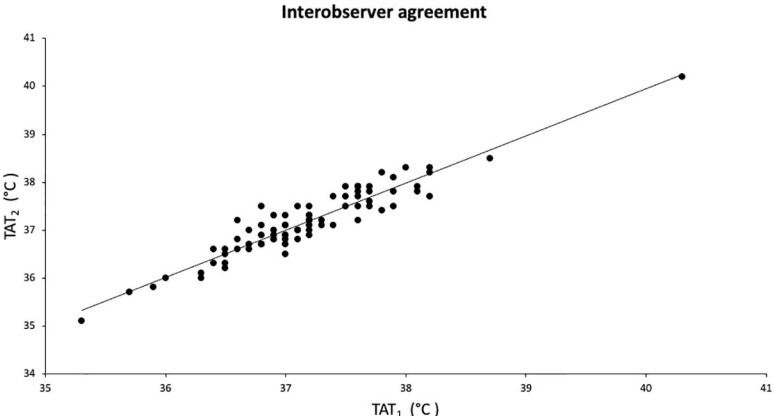

**Fig 4. Scatterplot interobserver agreement.** Scatter plot for body temperature data. Correlation between body temperature data obtained by observer 1 ($TAT_1$) and observer 2 ($TAT_2$) is illustrated.

authors found a low mean difference of 0.07˚C, but high LOA -1.48˚C to 1.62˚C, which corresponds fairly to the mean difference and LOA of our study.

Subgroup analysis showed that when patients were receiving norepinephrine (n = 41), the measurements of the test method deviated more from the reference method compared to when patients were not receiving norepinephrine. Authors of a previous study hypothesised that vasoactive medication could have influenced the accuracy of the TAT measurements [17]. However, they did not have enough patients to conduct a meaningful analysis of the impact of these factors (n = 21) [17]. Another study found that vasopressor use did not significantly increase bias in paediatric intensive care unit patients [18]. This may be explained because the sample size of the study was relatively small (n = 6) and included young children (median age 11.5 months) [18]. Based on our findings (n = 90), we are not able to recommend the use of the widely implemented TAT for intermittent body temperature measurements because of its poor agreement with BT, the reference method, in critically ill patients.

## Implications and generalizability

In an environment such as the ICU, where accurate and reliable temperature measurements are important to health care providers it is recommended to avoid the use of TAT and be aware and cautious about the accuracy and precision of its readings. This seems especially true in patients receiving vasoactive medication, however, research on the accuracy of the TAT in patients receiving vasoactive medication is scarce. Validation of our results in another cohort may strengthen our results and generalizability. However, this can be difficult to investigate in general wards or out-of-hospital clinics where patients do not have bladder catheters and where vasoactive medications are not used in daily practice. Someone might argue that it is expected that a sensory measured urinary bladder temperature would provide a different value than a transdermal measurement. The usefulness might still be considered if the measurement deviation could be systematically corrected. However, this seems not the case. The measurement deviations were completely unpredictable. This unpredictability and the degree to which the deviation manifests itself makes the thermometer in principle useless in the whole group of seriously ill patients.

## Strengths and limitations

Strengths of this study were that all measurements were performed by two independent researchers and that the sample size of our subgroup was relatively large compared to previous

studies. Before the start of this study, both observers received the same training programme and supervision, which has contributed to the strong correlation between the two observers. A strong positive correlation was found between two independent observers indicating that the user does not influence the accuracy of the measurements. This is in line with a study of McConnal et al. who reported acceptable interrater reliability with trained researchers [19].

There are also several limitations to this study. First, we have a small number of measurements in patients who were severely hypothermic or had febrile range temperatures. Therefore, statements about the accuracy of the TAT under febrile or deep hypothermal conditionals are not provided. Further studies with larger sample sizes including hypothermic and patients with febrile range temperatures might further increase generalizability of this study. Second, we used the BT as reference standard to measure core temperature instead of the PA catheter which is considered the golden standard. PA catheters however are invasive and infrequently used in clinical practice [2, 20]. Conversely, bladder thermistors are less invasive and provide continuous readings that are essentially identical to intravascular thermometers over a wide range of temperatures [21]. The inaccuracies caused by this core temperature measurement site may have contributed somewhat to the results. Third, the observers were not blinded for each other's measurements. However, as temperature is an objective measurement, we believe this would not have changed the results. Last, we assessed if patients received vasoactive medication at the time of the measurements. We did not assess if patients received vasoactive medication just before or after the measurements, this could have influenced our results. However, as most patients received continuous vasoactive medication during their ICU stay, it is unlikely that this changed short before or after the measurements.

## Conclusion

The TAT exceeds the clinical acceptable margins considerably in critically ill patients, especially in patients receiving norepinephrine. Therefore, this method is not sufficiently accurate for clinical use in critically ill patients.

## Supporting information

**S1 Table. Deviation of temperature measurements using different acceptable margins.** $TAT_1$: temporal artery measurement by observer 1; $TAT_2$: temporal artery measurement by observer 2.
(DOCX)

**S1 Data.**
(XLSX)

## Acknowledgments

We would like to thank all researchers and coordinators from the SICS Study Group. Iwan C. C. van der Horst, MD, PhD; Frederik Keus, MD, PhD; Jacqueline Koeze, MD; Geert Koster, MD; and Marisa Onrust. And also, our nursing students for their devoted involvement with patient inclusions: Hannah Klunder, Rianda D. Paapst.

## Author Contributions

**Conceptualization:** Eline G. M. Cox, Willem Dieperink, Renske Wiersema.

**Data curation:** Eline G. M. Cox, Willem Dieperink.

**Formal analysis:** Eline G. M. Cox, Frank Doesburg.

**Methodology:** Eline G. M. Cox, Willem Dieperink, Renske Wiersema, Ingeborg C. van der Meulen, Wolter Paans.

**Supervision:** Willem Dieperink, Renske Wiersema.

**Writing – original draft:** Eline G. M. Cox.

**Writing – review & editing:** Renske Wiersema, Frank Doesburg, Ingeborg C. van der Meulen, Wolter Paans.

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
