## [Decision Letter · Decision Letter 0]

9 Sep 2020

PONE-D-20-24690

Temporal auricular temperature measurements versus bladder temperature in critically ill patients, a prospective observational study

PLOS ONE

Dear Dr. Cox,

Thank you for submitting your manuscript to PLOS ONE. After careful consideration, we feel that it has merit but does not fully meet PLOS ONE’s publication criteria as it currently stands. Therefore, we invite you to submit a revised version of the manuscript that addresses the points raised during the review process.

We look forward to receiving your revised manuscript.

Kind regards,

Matthew John Meyer, M.D.

Academic Editor

PLOS ONE

Additional Editor Comments:

Thank you for your submission and your hard work on this study and manuscript.

Please answer our reviewer's questions as well as these few questions and re-submit for further consideration:

1) Were the examiners blinded (TAT 1 and TAT 2) to each other and to the bladder temperature?

2) What is “moisture” included alongside of vasopressors and heating in table?

3) Did you compare the absolute values of the differences between the two methods of temperature evaluation? It seems the mean is close to zero but the SD are quite large. And the absolute magnitude of the deviation is at least as meaningful as the direction (too hot / too cold) of the deviation. If not, I would like to see it run as a post-hoc analysis.

Journal Requirements:

2. Please describe in your methods section how capacity to consent was determined for the participants in this study.

Reviewers' comments:

Reviewer's Responses to Questions

**Comments to the Author**

1. Is the manuscript technically sound, and do the data support the conclusions?

Reviewer #1: Yes

2. Has the statistical analysis been performed appropriately and rigorously? 

Reviewer #1: Yes

3. Have the authors made all data underlying the findings in their manuscript fully available?

Reviewer #1: Yes

4. Is the manuscript presented in an intelligible fashion and written in standard English?

Reviewer #1: Yes

5. Review Comments to the Author

Reviewer #1: Peer Review for manuscript titled: Temporal auricular temperature measurements versus bladder temperature in critically ill patients, a prospective observational study

Thank you for submitting this manuscript for the method-comparison study aimed to evaluate the degree and limits of agreement between temporal artery temperature (TAT) and bladder temperature (BT) measurements. This study found good agreement between these thermometry methods with a mean difference (bias) of 0.12° C within the a priori set clinically meaningful threshold of < 0.5° C and the poor limit of agreement (precision) with a SD ± 0.61° C (95% CI -1.08°C, 1.32°C). This was a well-designed and executed study that addresses a gap in the method-comparison literature regarding thermometry methods used in hospitalized and critically ill patients, especially when non-invasive methods are preferred to reduce device-associated risk of harm to patients. I have primarily minor recommendations for the authors regarding the manuscript regarding clarifications of the methods, design, and result figures requiring attention prior to acceptance for publication.

1. Terminology/definitions throughout the manuscript: 1) The title and a few times in the manuscript, temporal auricular temperature or “auricular” is used (title, line 116, figure 2). I am not sure if this is a typo or misuse of the name of the thermometry method. Recommend using what is used in the literature, “Temporal Artery Temperature” when TAT is referred to. 2) TAT is obtained via a non-invasive method and is a peripheral temperature. Urinary bladder temperatures are obtained via an invasive method and is considered a core temperature (very good agreement with the gold standard pulmonary artery blood temperature obtained from a PA catheter with a thermistor). It is not customary to refer to BTs as semi-invasive than PA catheter rather these are either invasive or non-invasive.

2. Study design: this is a method-comparison study design. You reference Bland & Altman (1999) –they also have more current design/analysis publications on method-comparison studies.

3. Line 86: Recommend adding a statement with the rationale for conducting the subgroup analysis comparing vasopressor to non-vasopressor use.

4. Methods: 1) Was TBI considered an exclusion criteria? It is not listed in the methods section, yet was referred to as a reason for exclusion in the Results section. 2) Measurement: Excellent that you were able to include 2 observers to test for inter-rater reliability issues. Check for a typo in line 113 (intra-rater vs inter-rater). 3) Consider adding a description of how the TAT is obtained since there are more than one procedure in the literature (temporal artery forehead scan vs temporal artery forehead along face to behind ear scan techniques –see Lawson L, Bridges EJ, et al (2007) and Carroll E, et al (2004).

5. Results: 1) Although you collected 2 sets of data per patient (TAT1 and TAT2) for the inter-rater reliability test, it is unclear if there is a need to present the data in Figure 2 (scatter plots of correlations –which is not relevant to inter-rater reliability). The correlation result of r = 0.94 analyzed with the Pearson Correlation Coefficient demonstrates excellent inter-rater reliability.

2) Figure 3: Bland Altman plots: Although described in the legion, it would be preferred to have the LOA values in the figure aside the dashed lines.

6. Discussion: 1) Line 217-220: typo and consider use of standard reference for “mean difference” (bias) as the quantified degree of agreement and “limits of agreement” (precision) as SD and the 95% LOAs. 2) Line 240 –this statement re: “…although this was not statistically significant” was not clear to me. 3) Line 266-270: recommend using hypothermic and febrile range temperatures. Want to not confuse hyperthermic conditions with fever/febrile conditions.

7. References: Recommend a review of literature for more current references

8. General grammar/typo/etc:

a. Line 71-72: “shock resistant” is not clear –is this a shock state of the patient or a device characteristic?

b. Line 144: clinically acceptable

c. Line 167 –I believe you mean perspiring rather than transpiring.

d. Line 185-186: reword this important result statement to ensure that it is clear that the proportion of measurement differences that were ≥ 0.5°C was 36%. Same recommendation for Line 190-191.

6. PLOS authors have the option to publish the peer review history of their article (what does this mean?). If published, this will include your full peer review and any attached files.

Reviewer #1: **Yes: **Hildy M. Schell-Chaple

---

## [Author Response · Author response to Decision Letter 0]

28 Sep 2020

Editor:

1) Were the examiners blinded (TAT 1 and TAT 2) to each other and to the bladder temperature?

Author response: Dear Editor, thank you very much for your comments. The examiners were not blinded to each other nor to the bladder temperature. We have now added this as a limitation in the discussion. 

2) What is “moisture” included alongside of vasopressors and heating in table?

Author response: We have changed the phrase ‘moisture’ to ‘moist skin’ as this was indeed unclear. With moist skin we indicate a sweaty skin, signs which may be present in sepsis and fever. 

3) Did you compare the absolute values of the differences between the two methods of temperature evaluation? It seems the mean is close to zero but the SD are quite large. And the absolute magnitude of the deviation is at least as meaningful as the direction (too hot / too cold) of the deviation. If not, I would like to see it run as a post-hoc analysis.

Author response: Thank you for pointing at this. We have compared this in the results section, but this was not clear enough. We have now added an extra sentence to clarify this finding. 

Journal Requirements:

2. Please describe in your methods section how capacity to consent was determined for the participants in this study.

Author response: We have amended the manuscript and have added all above mentioned points to the manuscript.

Author response: There are no ethical or legal restrictions on sharing the de-identified data set. We have uploaded a minimal and anonymized data set as Supporting Information file.

Reviewers' comments:

Comments to the Author

Reviewer #1: Peer Review for manuscript titled: Temporal auricular temperature measurements versus bladder temperature in critically ill patients, a prospective observational study

Thank you for submitting this manuscript for the method-comparison study aimed to evaluate the degree and limits of agreement between temporal artery temperature (TAT) and bladder temperature (BT) measurements. This study found good agreement between these thermometry methods with a mean difference (bias) of 0.12° C within the a priori set clinically meaningful threshold of < 0.5° C and the poor limit of agreement (precision) with a SD ± 0.61° C (95% CI -1.08°C, 1.32°C). This was a well-designed and executed study that addresses a gap in the method-comparison literature regarding thermometry methods used in hospitalized and critically ill patients, especially when non-invasive methods are preferred to reduce device-associated risk of harm to patients. I have primarily minor recommendations for the authors regarding the manuscript regarding clarifications of the methods, design, and result figures requiring attention prior to acceptance for publication.

Author response: Thank you for your comments and reviewing this manuscript. 

1. Terminology/definitions throughout the manuscript: 1) The title and a few times in the manuscript, temporal auricular temperature or “auricular” is used (title, line 116, figure 2). I am not sure if this is a typo or misuse of the name of the thermometry method. Recommend using what is used in the literature, “Temporal Artery Temperature” when TAT is referred to. 2) TAT is obtained via a non-invasive method and is a peripheral temperature. Urinary bladder temperatures are obtained via an invasive method and is considered a core temperature (very good agreement with the gold standard pulmonary artery blood temperature obtained from a PA catheter with a thermistor). It is not customary to refer to BTs as semi-invasive than PA catheter rather these are either invasive or non-invasive. 

Author response: Thank you very much for pointing at this inconsistency. We have now changed this throughout the whole manuscript. 

2. Study design: this is a method-comparison study design. You reference Bland & Altman (1999) –they also have more current design/analysis publications on method-comparison studies.

Author response: Indeed, thank you for pointing this out. We will update our references. 

3. Line 86: Recommend adding a statement with the rationale for conducting the subgroup analysis comparing vasopressor to non-vasopressor use.

Author response: We have added a sentence to line 79 to clarify the rationale for conducting the subgroup analysis.

4. Methods: 1) Was TBI considered an exclusion criteria? It is not listed in the methods section, yet was referred to as a reason for exclusion in the Results section. 2) Measurement: Excellent that you were able to include 2 observers to test for inter-rater reliability issues. Check for a typo in line 113 (intra-rater vs inter-rater). 3) Consider adding a description of how the TAT is obtained since there are more than one procedure in the literature (temporal artery forehead scan vs temporal artery forehead along face to behind ear scan techniques –see Lawson L, Bridges EJ, et al (2007) and Carroll E, et al (2004).

Author response: 1) The accessibility of the places for measurement was mentioned in the methods section (line 99), which can be inaccessible due to TBI. 2) We have now changed this. 3) We have added a sentence (line 112-114) to clarify this in the methods section under measurement procedure. 

5. Results: 1) Although you collected 2 sets of data per patient (TAT1 and TAT2) for the inter-rater reliability test, it is unclear if there is a need to present the data in Figure 2 (scatter plots of correlations –which is not relevant to inter-rater reliability). The correlation result of r = 0.94 analyzed with the Pearson Correlation Coefficient demonstrates excellent inter-rater reliability. 2) Figure 3: Bland Altman plots: Although described in the legion, it would be preferred to have the LOA values in the figure aside the dashed lines.

Author response: 1) We believe that such scatter plots may aid the reader with interpreting our data and results. We agree however with you comment and we leave it up to the editor whether we will remove this figure. 2) We have now added the LOA values to the Bland Altman plots. 

6. Discussion: 1) Line 217-220: typo and consider use of standard reference for “mean difference” (bias) as the quantified degree of agreement and “limits of agreement” (precision) as SD and the 95% LOAs. 2) Line 240 –this statement re: “…although this was not statistically significant” was not clear to me. 3) Line 266-270: recommend using hypothermic and febrile range temperatures. Want to not confuse hyperthermic conditions with fever/febrile conditions.

Author response: 1) We have changed this in the discussion section (line 242-243). 2) This was indeed not clear; we have now removed this sentence from the manuscript. 3) We have changed this in the manuscript.

7. References: Recommend a review of literature for more current references

Author response: We have added and updated some references.

8. General grammar/typo/etc:

a. Line 71-72: “shock resistant” is not clear –is this a shock state of the patient or a device characteristic?

b. Line 144: clinically acceptable

c. Line 167 –I believe you mean perspiring rather than transpiring.

d. Line 185-186: reword this important result statement to ensure that it is clear that the proportion of measurement differences that were ≥ 0.5°C was 36%. Same recommendation for Line 190-191.

Author response: We have changed all grammar and typo’s according to the suggestions.

---

## [Decision Letter · Decision Letter 1]

22 Oct 2020

Temporal artery temperature measurements versus bladder temperature in critically ill patients, a prospective observational study

PONE-D-20-24690R1

Dear Dr. Cox,

We’re pleased to inform you that your manuscript has been judged scientifically suitable for publication and will be formally accepted for publication once it meets all outstanding technical requirements.

Kind regards,

Yu Ru Kou, PhD

Academic Editor

PLOS ONE

Additional Editor Comments (optional):

Reviewers' comments:

Reviewer's Responses to Questions

**Comments to the Author**

1. If the authors have adequately addressed your comments raised in a previous round of review and you feel that this manuscript is now acceptable for publication, you may indicate that here to bypass the “Comments to the Author” section, enter your conflict of interest statement in the “Confidential to Editor” section, and submit your "Accept" recommendation.

Reviewer #1: All comments have been addressed

2. Is the manuscript technically sound, and do the data support the conclusions?

Reviewer #1: Yes

3. Has the statistical analysis been performed appropriately and rigorously? 

Reviewer #1: Yes

4. Have the authors made all data underlying the findings in their manuscript fully available?

Reviewer #1: Yes

5. Is the manuscript presented in an intelligible fashion and written in standard English?

Reviewer #1: Yes

6. Review Comments to the Author

Reviewer #1: Thank you for revising the manuscript.

7. PLOS authors have the option to publish the peer review history of their article (what does this mean?). If published, this will include your full peer review and any attached files.

Reviewer #1: **Yes: **Hildy Schell-Chaple, PhD, RN, CCNS

---

## [Editor Report · Acceptance letter]

26 Oct 2020

PONE-D-20-24690R1 

Temporal artery temperature measurements versus bladder temperature in critically ill patients, a prospective observational study 

Dear Dr. Cox:

I'm pleased to inform you that your manuscript has been deemed suitable for publication in PLOS ONE. Congratulations! Your manuscript is now with our production department. 

Kind regards, 

on behalf of

Dr. Yu Ru Kou 

Academic Editor

PLOS ONE